# Allosteric communication in class A β-lactamases occurs via cooperative coupling of loop dynamics

Ioannis Galdadas[1†], Shen Qu[2†], Ana Sofia F Oliveira[3], Edgar Olehnovics[2], Andrew R Mack[4,5], Maria F Mojica[4,6], Pratul K Agarwal[7], Catherine L Tooke[8], Francesco Luigi Gervasio[1,9,10], James Spencer[8], Robert A Bonomo[4,5,6,11,12,13,14], Adrian J Mulholland[3]*, Shozeb Haider[2]*

[1]University College London, Department of Chemistry, London, United Kingdom; [2]University College London School of Pharmacy, Pharmaceutical and Biological Chemistry, London, United Kingdom; [3]University of Bristol, Centre for Computational Chemistry, School of Chemistry, Bristol, United Kingdom; [4]Veterans Affairs Northeast Ohio Healthcare System, Research Service, Cleveland, United States; [5]Case Western Reserve University, Department of Molecular Biology and Microbiology, Cleveland, United States; [6]Case Western Reserve University, Department of Infectious Diseases, School of Medicine, Cleveland, United States; [7]Department of Physiological Sciences and High-Performance Computing Center, Oklahoma State University, Stillwater, United States; [8]University of Bristol, School of Cellular and Molecular Medicine, Bristol, United Kingdom; [9]University College London, Institute of Structural and Molecular Biology, London, United Kingdom; [10]University of Geneva, Pharmaceutical Sciences, Geneva, Switzerland; [11]Case Western Reserve University, Department of Biochemistry, Cleveland, United States; [12]Case Western Reserve University, Department of Pharmacology, Cleveland, United States; [13]Case Western Reserve University, Department of Proteomics and Bioinformatics, Cleveland, United States; [14]CWRU-Cleveland VAMC Center for Antimicrobial Resistance and Epidemiology (Case VA CARES), Cleveland, United States

*For correspondence:
adrian.mulholland@bristol.ac.uk (AJM);
shozeb.haider@ucl.ac.uk (SH)

†These authors contributed equally to this work

**Abstract** Understanding allostery in enzymes and tools to identify it offer promising alternative strategies to inhibitor development. Through a combination of equilibrium and nonequilibrium molecular dynamics simulations, we identify allosteric effects and communication pathways in two prototypical class A β-lactamases, TEM-1 and KPC-2, which are important determinants of antibiotic resistance. The nonequilibrium simulations reveal pathways of communication operating over distances of 30 Å or more. Propagation of the signal occurs through cooperative coupling of loop dynamics. Notably, 50% or more of clinically relevant amino acid substitutions map onto the identified signal transduction pathways. This suggests that clinically important variation may affect, or be driven by, differences in allosteric behavior, providing a mechanism by which amino acid substitutions may affect the relationship between spectrum of activity, catalytic turnover, and potential allosteric behavior in this clinically important enzyme family. Simulations of the type presented here will help in identifying and analyzing such differences.

**eLife digest** Antibiotics are crucial drugs for treating and preventing bacterial infections, but some bacteria are evolving ways to resist their effects. This 'antibiotic resistance' threatens lives and livelihoods worldwide. β-lactam antibiotics, like penicillin, are some of the most commonly used, but some bacteria can now make enzymes called β-lactamases, which destroy these antibiotics. Dozens of different types of β-lactamases now exist, each with different properties. Two of the most medically important are TEM-1 and KPC-2.

One way to counteract β-lactamases is with drugs called inhibitors that stop the activity of these enzymes. The approved β-lactamase inhibitors work by blocking the part of the enzyme that binds and destroys antibiotics, known as the 'active site'. The β-lactamases have evolved, some of which have the ability to resist the effects of known inhibitors. It is possible that targeting parts of β-lactamases far from the active site, known as 'allosteric sites', might get around these new bacterial defences. A molecule that binds to an allosteric site might alter the enzyme's shape, or restrict its movement, making it unable to do its job.

Galdadas, Qu et al. used simulations to understand how molecules binding at allosteric sites affect enzyme movement. The experiments examined the structures of both TEM-1 and KPC-2, looking at how their shapes changed as molecules were removed from the allosteric site. This revealed how the allosteric sites and the active site are linked together. When molecules were taken out of the allosteric sites, they triggered ripples of shape change that travelled via loop-like structures across the surface of the enzyme. These loops contain over half of the known differences between the different types of β-lactamases, suggesting mutations here may be responsible for changing which antibiotics each enzyme can destroy. In other words, changes in the 'ripples' may be related to the ability of the enzymes to resist particular antibiotics.

Understanding how changes in one part of a β-lactamase enzyme reach the active site could help in the design of new inhibitors. It might also help to explain how β-lactamases evolve new properties. Further work could show why different enzymes are more or less active against different antibiotics.

## Introduction

The rise in antimicrobial resistance (AMR) is a growing global public health crisis (*Centers for Disease Control and Prevention (U.S.), 2019*). As AMR has continued to spread and many antimicrobial agents have become ineffective against previously susceptible organisms, the World Health Organization recently projected that AMR could result in up to 10 million deaths annually by 2050 (*Interagency Coordination Group on Antimicrobial Resistance, 2019*). The problem of AMR is particularly urgent given the alarming proliferation of antibiotic resistance in bacteria; pathogens associated with both community-acquired and healthcare-associated infections are increasingly resistant to first-line and even reserve agents (*Lythell et al., 2020*). This not only poses a serious challenge obstacle in fighting common and severe bacterial infections, but also reduces the viability and increases the risks of interventions such as orthopedic surgery and also threatens new antibiotics coming to the market (*Bush and Page, 2017*). AMR risks negating a century of progress in medicine made possible by the ability to effectively treat bacterial infections.

In spite of the advances in the field of antimicrobial chemotherapy, the efficacy, safety, chemical malleability, and versatility of β-lactams make them the most prescribed class of antibiotics (*Tooke et al., 2019*). Their cumulative use exceeds 65% of all injectable antibiotics in the United States (*Bush and Bradford, 2016*). β-Lactam antibiotics work by inhibiting penicillin binding proteins (PBPs), a group of enzymes that catalyze transpeptidation and transglycosylation reactions that occur during the bacterial cell wall biosynthesis (*Tooke et al., 2019*). A damaged cell wall results in loss of cell shape, osmotic destabilization, and is detrimental for bacterial survival in a hypertonic and hostile environment (*Bonomo, 2017*). Of the four primary mechanisms by which bacteria resist β-lactam antibiotics, the most common and important mechanism of resistance in Gram-negative bacteria, including common pathogens such as *Escherichia coli* and *Klebsiella pneumoniae*, is the expression of β-lactamase enzymes (*Tooke et al., 2019*). These enzymes hydrolyze the amide bond in the β-lactam ring, resulting in a product that is incapable of inhibiting PBPs (*Palzkill, 2018*).

The Ambler system of classifying β-lactamase enzymes categorizes them, based on amino acid sequence homology, into classes A, B, C, and D (*Ambler, 1980*; *Bush and Jacoby, 2010*). While β-lactamases of classes A, C, and D are serine hydrolases, class B enzymes are metalloenzymes that have one or more zinc ions at the active site (*Palzkill, 2013*). Class A enzymes are the most widely distributed and intensively studied of all β-lactamases (*Tooke et al., 2019*). The hydrolytic mechanism in class A (*Figure 1—figure supplement 1*), revealed by experiments and QM/MM modeling, is initiated by reversible binding of the antibiotic in the active site of the enzyme (formation of the Michaelis complex). This is followed by nucleophilic attack of the catalytic serine (Ser70) on the carbonyl carbon of the β-lactam ring, resulting in a high-energy acylated intermediate that quickly resolves, following protonation of the β-lactam nitrogen and cleavage of the C-N bond, to a lower energy covalent acyl-enzyme complex (*Chudyk et al., 2014*; *Hermann et al., 2003*; *Hermann et al., 2005*). Next, an activated water molecule attacks the covalent complex, leading to the subsequent hydrolysis of the bond between the β-lactam carbonyl and the serine oxygen, resulting in the regeneration of the active enzyme and release of the inactive β-lactam antibiotic (*Tooke et al., 2019*; *Bonomo, 2017*; *Palzkill, 2018*; *Chudyk et al., 2014*; *Fisher and Mobashery, 2009*; *Hermann et al., 2006*; *Hirvonen et al., 2019*; *Pan et al., 2017*).

TEM-1 is one of the most common plasmid-encoded β-lactamases in Gram-negative bacteria and is a model class A enzyme (*Brown et al., 2009*). It has a narrow spectrum of hydrolytic activity that is limited to penicillins and early generation cephalosporins; in contrast, its activity toward large, inflexible, broad-spectrum oxyiminocephalosporins such as the widely used antibiotic ceftazidime is poor (*Palzkill, 2018*). However, mutations in the *bla*_TEM-1 gene have led to amino acid modifications, which allow subsequent TEM-1 variants to hydrolyze broad-spectrum cephalosporins (so-called 'extended-spectrum' activity) or to avoid the action of mechanism-based inhibitors such as clavulanate that are used in combination with β-lactams to treat β-lactamase producing organisms (*Brown et al., 2009*). Another class A enzyme, KPC-2 (*K. pneumoniae* carbapenemase-2), encoded by the *bla*_KPC-2 gene is an extremely versatile β-lactamase (*Queenan et al., 2004*) with a broad spectrum of substrates that includes penicillins, cephamycins, and, importantly, carbapenems (*Queenan et al., 2004*; *Yigit et al., 2003*). Currently, predominant strains of *K. pneumoniae* and other Enterobacterales continue to be identified as responsible for outbreaks internationally. Continued dissemination of KPC makes this one of the β-lactamases of most immediate clinical importance and a key target for inhibitor development.

The structure and activity of class A β-lactamases have been well studied (*Palzkill, 2018*; *Papp-Wallace et al., 2012*; *Salverda et al., 2010*). In spite of sequence differences, class A β-lactamases share the same structural architecture (*Philippon et al., 2016*), as evident from the present 47 structures of TEM-1 and 38 structures of KPC-2, or their engineered variants, deposited in the Protein Data Bank (PDB) at the time of this writing. However, despite the wide variety of substrates that TEM-1 and KPC-2 can hydrolyze, their structures are quite rigid. The average mean order parameter, $S^2$, as calculated from NMR experiments for TEM-1, is between 0.81 and 0.94, and almost all class A β-lactamases are conformationally identical (*Gobeil et al., 2019*; *Morin and Gagné, 2009*; *Savard and Gagné, 2006*). Loops (e.g. active site loops) play a crucial role in the activity of many enzymes (*Liao et al., 2018*), including β-lactamases. There is increasing evidence that active site conformations may be influenced by distal loops, connected, for example, through active closure and desolvation, and potentially via networks of coupled motions (*Liao et al., 2018*; *Agarwal, 2019*; *Bunzel et al., 2020*; *Bunzel et al., 2021*). The active sites of TEM-1 and KPC-2 are surrounded by three loops: (a) the Ω-loop (residues 172–179), (b) the loop between α3 and α4 helices, in which a highly conserved aromatic amino acid is present at position 105, and (c) the hinge region, which lies opposite to the Ω-loop and contains the α11 helix turn (*Figure 1*, *Figure 1—figure supplement 2*). Two highly conserved residues, Glu166 and Asn170, which are essential for catalysis, influence the conformation of the Ω-loop (*Banerjee et al., 1998*). The conformational dynamics of these loops play an important role in enzyme activity and are probably modulated by evolution (*Pan et al., 2017*; *Banerjee et al., 1998*; *Escobar et al., 1994*; *Guillaume et al., 1997*; *Leung et al., 1994*; *Zawadzke et al., 1996*). For example, we have recently found that differences in the spectrum of activity between KPC-2 and KPC-4 are due to changes in loop behavior (*Tooke et al., 2021*).

There has been extensive discussion about the possible contribution of protein dynamics to enzyme catalysis (*Glowacki et al., 2012*; *Kamerlin and Warshel, 2010*; *Luk et al., 2013*; *Singh et al., 2015*). In some enzymes, conformational changes have been identified as necessary in

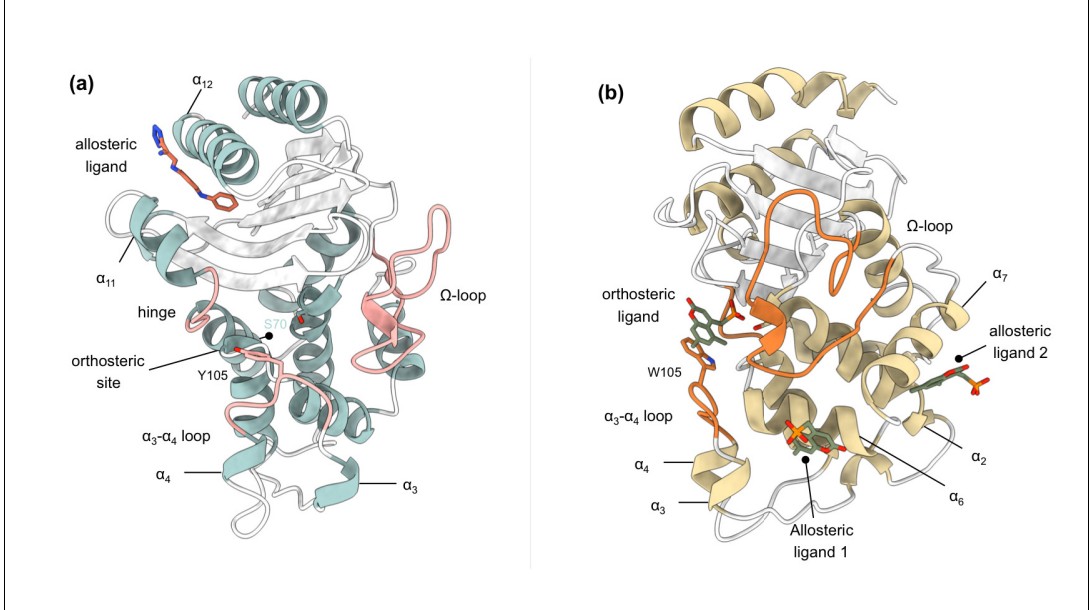

**Figure 1.** Crystal structures of (**a**) TEM-1 (PDB id 1PZP) and (**b**) KPC-2 (PDB id 6D18) β-lactamases in complex with ligands bound to allosteric and the orthosteric sites. The helices around the allosteric binding sites and the loops that define the orthosteric binding site are highlighted. In case of KPC-2, allosteric ligand 2 is the site investigated here. See Table S1 for structural nomenclature.

The online version of this article includes the following figure supplement(s) for figure 1:

**Figure supplement 1.** Catalytic cycle of a class A β-lactamase illustrated on the core structure of penicillins.

**Figure supplement 2.** Naming of the loops based on the secondary structure it connects.

preparing the system for reaction (*Liao et al., 2018*; *Agarwal, 2019*). Several simulation studies, including long timescale and enhanced sampling molecular dynamics (MD) simulations and QM/MM simulations of reactions, have been reported for TEM-1 and KPC-2 β-lactamases (*Chudyk et al., 2014*; *Hirvonen et al., 2019*; *Bowman et al., 2015*; *Galdadas et al., 2018*; *Hart et al., 2016*; *Tooke et al., 2021*). MD simulations have explored cryptic pocket formation (*Hart et al., 2016*), studied protein-ligand interactions (*Fisette et al., 2012*), predicted antibiotic resistance (*Chudyk et al., 2014*; *Hirvonen et al., 2019*; *Galdadas et al., 2018*), explained the effects of mutations on enzyme specificities (*Zaccolo and Gherardi, 1999*), and investigated conserved hydrophobic networks (*Galdadas et al., 2018*). It remains a challenge to directly link conformational heterogeneity and function.

Understanding conformational behavior is relevant to β-lactamase inhibition as well as catalytic mechanism. For organisms producing class A β-lactamases, co-administration of susceptible β-lactams with mechanism-based covalent inhibitors (e.g. clavulanate) represents a proven therapeutic strategy and has successfully extended the useful lifetime of penicillins in particular (*Fritz et al., 2018*; *Drawz and Bonomo, 2010*). However, while the mechanism of direct inhibition by covalently bound inhibitors is well established (*Fritz et al., 2018*), the possibility of exploiting sites remote from the active center in allosteric inhibition strategies is less well explored, and where this has been achieved (*Horn and Shoichet, 2004*; *Pemberton et al., 2019*; *Hart et al., 2016*) the structural changes occurring as a result of ligand binding or unbinding to allosteric sites and the relay of structural communication that leads to inhibition are not well understood. The conformational rearrangements that take place upon ligand (un)binding in allosteric sites and their potential connection to the β-lactamase active site are the focus of this study.

Here, we employ a combination of equilibrium and nonequilibrium MD simulations to identify and study the response of two class A β-lactamases, TEM-1 and KPC-2, to the (un)binding of ligands at sites distant from the active site. Nonequilibrium simulations applying the Kubo-Onsager approach (*Ciccotti and Ferrario, 2016*; *Ciccotti et al., 1979*) are emerging as an effective way to characterize conformational changes and communication networks in proteins (*Abreu et al., 2020*; *Damas et al., 2011*; *Oliveira et al., 2019a*; *Oliveira et al., 2019b*).

To the best of our knowledge, this is the first application of this nonequilibrium MD approach to study enzymes. We study β-lactamases, whose ultrafast turnover rates can approach the diffusion limits for natural substrates ($\sim10^7$–$10^8$ M$^{-1}$s$^{-1}$) (*Fisher and Mobashery, 2009*). We perform 10 µs of equilibrium MD simulations of TEM-1 and KPC-2, with and without ligands present in their allosteric binding sites. These simulations identify conformational changes in the highly dynamic loops that shape the active site and structurally characterize the dynamics of the formation and dissolution of the allosteric pocket. We also carry out an extensive complementary set of 1600 short nonequilibrium MD simulations (a total of 8 µs of accumulated time), which reveal the response of the enzyme to perturbation and identify pathways in the enzymes that connect the allosteric site to other parts of the protein. These simulations demonstrate direct communication between the allosteric sites and the active site. The results show that this combination of equilibrium and nonequilibrium MD simulations offers a powerful tool and a promising approach to identify allosteric communication networks in enzymes.

## Results

### Equilibrium simulations of Apo$_{EQ}$ and IB$_{EQ}$ states

To explore the conformational space of TEM-1 and KPC-2 in the Apo$_{EQ}$ (no ligand) and IB$_{EQ}$ (inhibitor-bound) states, we started by running a set of equilibrium simulations (20 replicas of 250 ns each) that resulted in 5 µs of accumulated simulation time per system. Conformational changes during the simulations were assessed using their Cα root mean-square deviation (RMSD) profiles (*Figure 2—figure supplement 2*). The simulated systems were considered equilibrated beyond 50 ns as shown by RMSD convergence. In each case, the proteins remained close to their initial conformation during the course of 250 ns (*Figure 2—figure supplement 2a*). The average RMSD for Apo$_{EQ}$ and IB$_{EQ}$ states were between 0.10 and 0.12 nm for all systems (*Figure 2—figure supplement 6*). The low RMSD values are consistent with previously published results, which have also shown class A β-lactamase enzymes to be largely rigid and conformationally stable when studied on long timescales and rarely divergent from the initial structure (*Gobeil et al., 2019*; *Galdadas et al., 2018*). Conventional RMSD fitting procedure using all Cα atoms failed to separate regions of high versus low mobility. To resolve such regions, we used a fraction (%) of the Cα atoms for the alignment. Beyond this fraction, there is a sharp increase in the RMSD value for the remainder of the Cα atoms (*Figure 2—figure supplement 2b*). At 80%, the core of TEM-1 could be superimposed to less than 0.064 and 0.074 nm for Apo$_{EQ}$ and IB$_{EQ}$ states, respectively (*Figure 2—figure supplement 2bi*).

In the KPC-2 Apo$_{EQ}$ state, the RMSD of 80% of the Cα atoms was below 0.060 nm, while the same subset of atoms had an RMSD below 0.066 nm in the IB$_{EQ}$ state (*Figure 2—figure supplement 2bii*). This 80% fraction of Cα atoms constitutes the core of the enzyme and did not show any divergence from the initial reference structure (*Figure 2—figure supplement 2c*). RMSD values for the remaining 20% of Cα atoms varied between 0.16 and 0.23 nm. This apparent rigidity is consistent with the experimental finding, based upon, for example, thermal melting experiments *Mehta et al., 2015*, that KPC-2 is more stable than many other class A β-lactamases such as TEM-1. Some large conformational changes were observed in all replicates; these involved changes in conformations of the loops that connect secondary structural elements (*Figure 2—figure supplement 3*). To further validate the stability of the two systems, we analyzed structural properties including the radius of gyration (Rg; Figure S5), solvent accessible surface area (SASA; *Figure 2—figure supplement 5*), and the secondary structure of each enzyme over the simulated time (*Figure 2—figure supplement 7*). The values for these properties are listed in *Figure 2—figure supplement 6*.

### Ligand-induced structural and dynamical changes

A ligand that binds to an allosteric site can control protein function by affecting the active site (*Laskowski et al., 2009*). This generally occurs by altering the conformational ensemble that the protein adopts (*Laskowski et al., 2009*; *Motlagh et al., 2014*). To probe how ligand binding to an allosteric site affects the dynamics of β-lactamases, we calculated the Cα root-mean-square fluctuation (RMSF) for both Apo$_{EQ}$ and IB$_{EQ}$ states. Higher RMSF values correspond to greater flexibility during the simulation. Although the Cα RMSF profiles for Apo$_{EQ}$ and IB$_{EQ}$ states are similar, indicating similar dynamics, there are some discernible differences (*Figure 2*).

In equilibrium simulations of TEM-1 and KPC-2, the hydrophobic core of the enzyme is stable and shows limited fluctuations. Most of the RMSF variance is observed in loops that connect secondary structural elements (*Figure 2*). In TEM-1 IB$_{EQ}$, higher fluctuations are observed predominantly in three distinct regions when compared with the Apo$_{EQ}$ enzyme; in the loops between helices α7 and α8 (residues 155–165), α9 and α10 (residues 196–200), and the hinge region including helix α11 (residues 213–224) (*Figure 2A*). The α11 and the α12 helices are part of a highly hydrophobic region that also constricts the allosteric pocket in all TEM-1 apo crystal structures. Binding of the ligand disrupts the hydrophobic interactions within this region, resulting in the opening of the allosteric pocket between helices α11 and α12 (*Horn and Shoichet, 2004*).

It should be noted that the starting Apo$_{EQ}$ structure of TEM-1 was generated from the IB crystal structure, by the removal of the ligand from the allosteric binding site. During the Apo$_{EQ}$ simulations, α12 helix behaves like a lid and closes over the empty, hydrophobic, allosteric binding site, and thus displays high RMSF at the C-terminal end of the enzyme. This conformational change recovers the structure of the apo crystal form, as observed, for example, in PDB id 1ZG4 (*Stec et al., 2005*), as reflected to the RMSD of ~0.07 nm after superposition of the structures. The rest of the loops displayed comparable fluctuations in both Apo$_{EQ}$ and IB$_{EQ}$ states.

The differences between the Apo$_{EQ}$ and IB$_{EQ}$ states were of similar magnitude in KPC-2. In KPC-2 IB$_{EQ}$, more extensive fluctuations than in Apo$_{EQ}$ were also observed in the loops between α7 and α8 (residues 156–166), the hinge region, around α11 (residues 214–225), and in the loop between β7 and β8 (residues 238–243) (*Figure 2b*). Conversely, fluctuations are slightly higher in the Apo$_{EQ}$ than IB$_{EQ}$ state in the loop leading into the Ω-loop from α7 helix (residues 156–166). Overall, however, RMSFs are similar in analogous regions of the IB$_{EQ}$ and Apo$_{EQ}$ states in both TEM-1 and KPC-2, highlighting the conservation of structural dynamics in class A β-lactamases. However, there were some fluctuations that were unique and limited to each enzyme (*Figure 2*).

In both TEM-1 and KPC-2 IB$_{EQ}$ states, interactions of the ligands in their respective allosteric binding sites contribute to enhanced fluctuations (i.e. larger than in the Apo forms) of the local structural elements (*Figure 3—figure supplement 1*). The sites in which the ligands bind are very different. In TEM-1, the binding site is deep and forms a hydrophobic cleft. The ligand penetrates to the core of the enzyme and is sandwiched between helices α11 and α12 (*Horn and Shoichet, 2004*). The FTA ligand remains tightly bound in the allosteric pocket throughout the simulations (*Figure 3—figure supplement 2*).

In KPC-2, the allosteric binding site is shallow and solvent-exposed even in the absence of the ligand. Although the distal end of the pocket is hydrophobic, there are some polar amino acids on the proximal surface (e.g. Arg83 and Gln86), which are exposed to the solvent. This shallow site forms a part of a larger pocket that is occluded by the side chain of Arg83 (α7 helix). In some of our IB$_{EQ}$ simulations, the Arg83 side chain rotates, leading to the opening of a larger hidden pocket. This enlarged space is now accessible to the ligand for exploring various interactions. The tumbling of GTV increases the fluctuations in the complex (*Figure 3—figure supplement 1c,d*), however, the ligand does not leave the binding site (*Figure 3—figure supplement 2*).

To further highlight the structural changes occurring as a result of ligand binding, positional C deviations were calculated between IB$_{EQ}$ and Apo$_{EQ}$ systems for the equilibrated part of the simulations (*Figure 3a,b*). The Cα deviation values plotted are an average between simulation taken by combining all trajectories from Apo$_{EQ}$ and IB$_{EQ}$ simulations for that particular system. This is one of the simplest approaches, which can determine residues undergoing largest structural rearrangements. The averaged Cα positional deviations are mapped onto the averaged Apo$_{EQ}$ structure to visualize the largest relative displacements in three dimensions (*Figure 3c,d*).

The hydrophobic cores of both TEM-1 and KPC-2 β-lactamase enzymes show little or no conformational change. The major differences between the Apo$_{EQ}$ and IB$_{EQ}$ states are in the loops connecting different secondary structure elements. In TEM-1, Cα deviations are observed in the loops between α4 and β5 (residues 112–116), α7 and α8 (residues 155–166), Ω-loop (residues 172–179), α9 and α10 (residues 196–200), hinge and α11 (residues 213–224), β7 and β8 (residues 238–243), and β9 and α12 (residues 267–272). There are some relatively minor deviations observed in loops β1-β2 (residues 51–55), β2-β3 (residues 61–65), α2-β4 (residues 86–93), α6-α7 (residues 143–144), β8-β9 (residues 252–258), and at the pivot of the α3 helix (residues 98–101). The hinge region and residues in helices α11 and α12 display the largest deviations. This is also in agreement with other experimental data that indicate the connection between the active site and the allosteric pocket

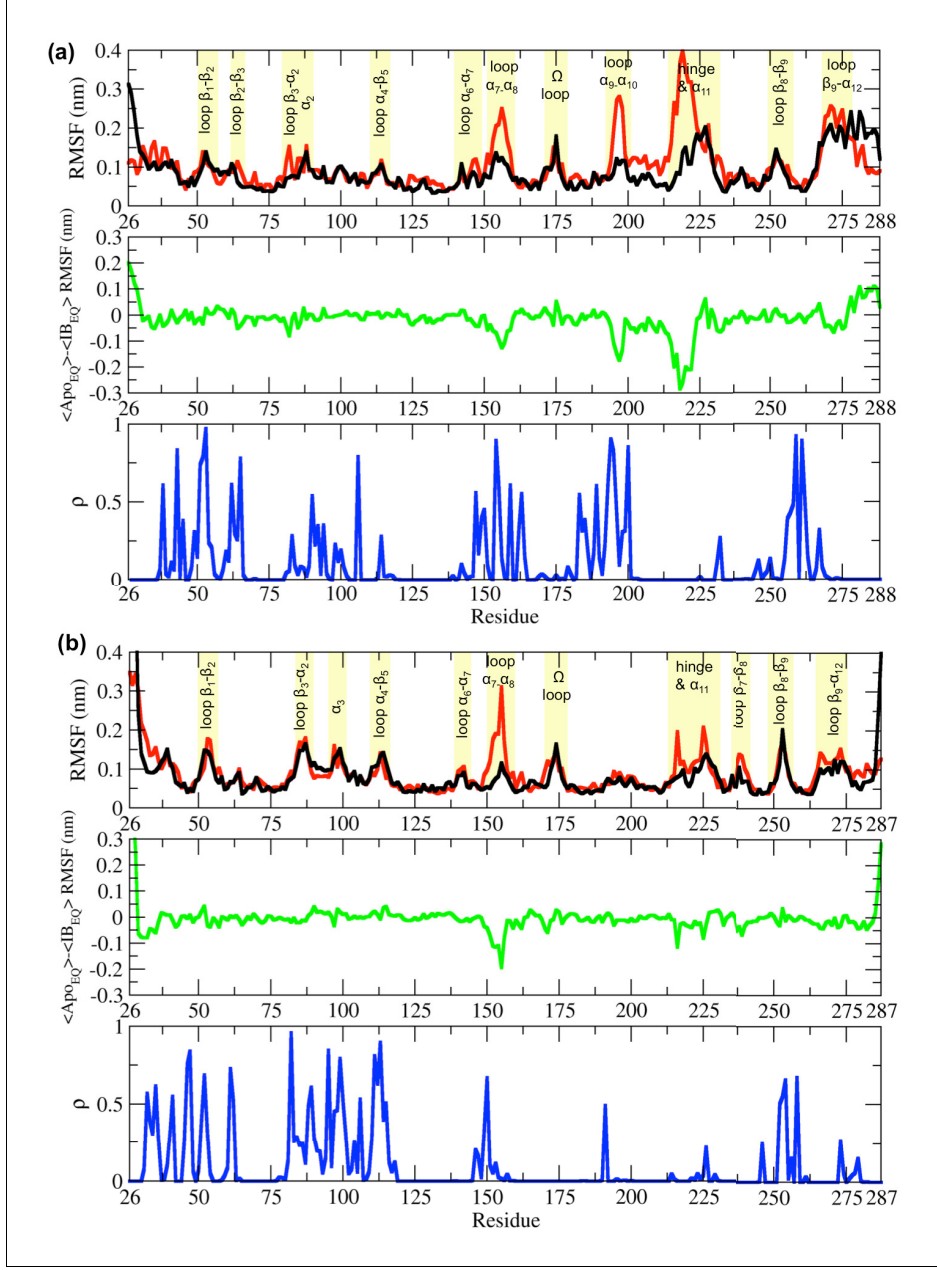

**Figure 2.** Root-mean-square fluctuation (RMSF) differences between the $Apo_{EQ}$ and $IB_{EQ}$ states of the (**a**) TEM-1 and (**b**) KPC-2 systems. The average change in RMSF in the $Apo_{EQ}$ (black), the $IB_{EQ}$ (red), the difference $Apo_{EQ}$-$IB_{EQ}$ (green), and the associated ρ value (blue) is illustrated. The ρ values were obtained by conducting a Student's t-test to compare $Apo_{EQ}$ and $IB_{EQ}$ systems and to assess the significance of the differences.

The online version of this article includes the following figure supplement(s) for figure 2:

**Figure supplement 1.** Schematic description of the long equilibrium (EQ) and short nonequilibrium (NE) simulations.

**Figure supplement 2.** Conformational drift measured by Cα root mean square deviation.

**Figure supplement 3.** Core Cα root-mean-square deviation (RMSD) superimposition from (a) TEM-1 and (b) KPC-2 $IB_{EQ}$ simulations.

**Figure supplement 4.** Time evolution of the radius of gyration (Rg) over the course of the 250 ns of each replicate.

**Figure supplement 5.** Solvent accessible surface area (SASA) over the course of the 250 ns of each replicate.

**Figure supplement 6.** Dynamical properties (root-mean-square deviation [RMSD], radius of gyration [Rg], and solvent accessible surface area [SASA]) used to assess structural stability of the systems over the course of the equilibrium simulation.

*Figure 2 continued on next page*

*Figure 2 continued*

**Figure supplement 7.** Probability to find each residue in a coil, helix, or strand over the course of the 250 ns of each replicate.

studied in TEM-1 in the presence of BLIP inhibitor, seems to be mostly due to hinge region motions (*Meneksedag et al., 2013*).

The structural dynamics observed in KPC-2 were slightly different from TEM-1. In KPC-2, prominent Cα deviations were observed in the loops between β1 and β2 (residues 51–55), α2 and β4 (residues 88–93), α4 and β5 (residues 114–116), α7 and α8 (residues 156–166), Ω-loop (residues 172–179), β7 and β8 (residues 238–243), β8 and β9 (residues 252–258), in the loop between β4 and α3 leading up to the proximal end of α3 (residues 94–102) and in the hinge/α11 helix (residues 214–225). There are some minor deviations observed in α1-β1 (residues 39–42) and β9-α12 (residues 266–270). The most important ligand-induced Cα deviation is observed in the loop connecting the α4 helix to the β5 strand (residues 114–116). The deviation of the α4-β5 loop together with the deviation observed in the loop between β4 and α3 leading into α3 helix (residues 96–102) has the potential to deform the α3 helix-turn-α4 helix. The β4-α3 and α4-β5 loops form the basal pivot joint of the α3 and α4 helices and maintain the correct positioning of this helix-turn-helix at the periphery of the enzyme active site. The correct positioning of this loop is important as Trp105 lies on this loop. Mutagenesis studies have shown that a highly conserved aromatic amino acid at position 105 in class A β-lactamases (Tyr105 in TEM-1, Trp105 in KPC-2) is located at the perimeter of the active site and plays a crucial role in ligand recognition via favorable stacking interactions with the β-lactam ring (*Papp-Wallace et al., 2010b*; *Doucet et al., 2004*). The aromatic side chain at position 105 coordinates the binding of substrates not only via stacking and edge-to-face interactions but by also adopting 'flipped-in' or 'flipped-out' conformations (*Galdadas et al., 2018*; *Papp-Wallace et al., 2010a*; *Papp-Wallace et al., 2010b*). This has been proposed based on the conformations observed in the available crystal structures and confirmed by enhanced sampling MD simulations (*Galdadas et al., 2018*; *Ke et al., 2012*). Any perturbation that alters the conformation of α3-turn-α4 helix or deforms the α3-α4 pivot region would prevent α3 and α4 helices from correctly shaping the active site of the enzyme. This would result in the aromatic residue at 105 partially detaching from the edge of the active site and being unable to stabilize the incoming substrate as required for efficient catalysis. This explains the loss of β-lactam resistance in strains expressing KPC variants at position 102 or 108, as established in the MIC experiments reported previously (*Galdadas et al., 2018*).

## Signal propagation from the allosteric site

To study signal propagation from the two allosteric sites, we ran 800 short nonequilibrium simulations, with a total sampling time of 4 μs for each system. The nonequilibrium simulations were initiated from regular intervals of the equilibrated part of the long $IB_{EQ}$ simulation, starting at the 50 ns time point (*Figure 2—figure supplement 1*). In each simulation, the ligand was removed from its binding site and the resulting system was further simulated for 5 ns. The response of the system to the perturbation was determined using the Kubo-Onsager approach developed by *Ciccotti and Ferrario, 2013*; *Ciccotti et al., 1979*; *Ciccotti and Ferrario, 2016*. In this approach, the time evolution of the conformational changes induced by ligand removal can be determined by comparing the $Apo_{NE}$ and $IB_{EQ}$ simulations at equivalent points in time. The subtraction method, applied to multiple pairs of trajectories, effectively removes noise arising from fluctuations of the systems and allows residues that are involved in signal propagation to be identified. The disappearance of the ligand from its binding site generates a temporary localized vacuum, against which there is an immediate structural and solvent response. As the simulation progresses, the cascading conformational changes in response to the perturbation (removal of ligand) show the route by which structural response is transmitted through the protein.

Video supplements: Signal propagation in TEM-1 and KPC-2 as a result of the perturbation (ligand removal) in the allosteric binding site. The disappearance of the ligand from its binding site generates a localized vacuum, against which there is an immediate structural response by the enzyme. As the simulation progresses, the cascading conformational changes in response to the

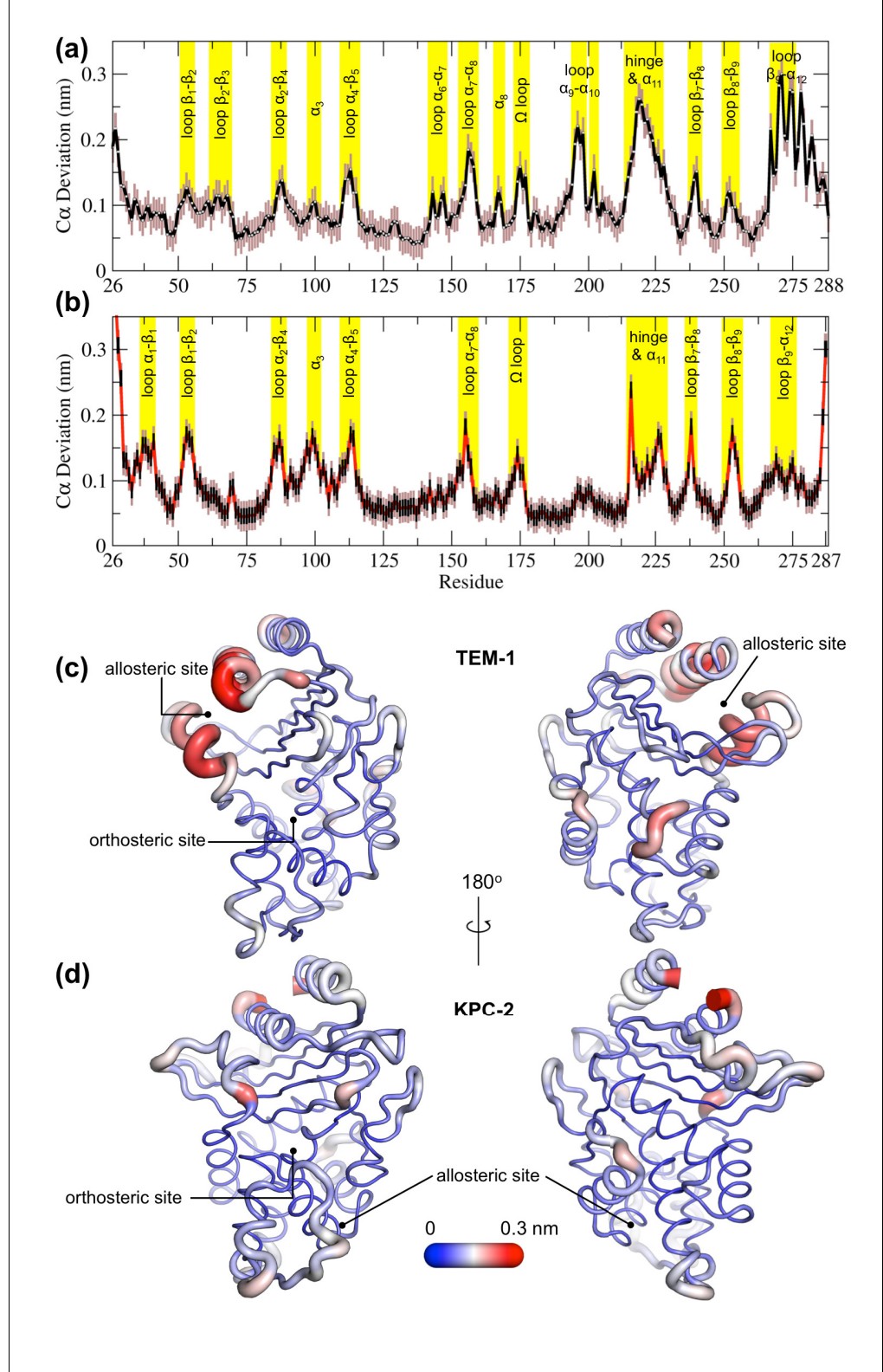

**Figure 3.** Average positional Cα deviations between the Apo$_{EQ}$ and IB$_{EQ}$ states of (**a**) TEM-1 and (**b**) KPC-2. Important structural motifs are highlighted and labeled on the plots. The brown vertical lines represent the standard deviation of the mean. The averaged Cα positional deviations mapped onto the averaged Apo$_{EQ}$ structures of (**c**) TEM-1 and (**d**) KPC-2 to visualize the largest relative displacements. The average deviation was

*Figure 3 continued on next page*

*Figure 3 continued*

determined from a combination of all 20 Apo$_{EQ}$ and 20 IB$_{EQ}$ trajectories. The thickness of the cartoon corresponds to the Cα deviation.

The online version of this article includes the following figure supplement(s) for figure 3:

**Figure supplement 1.** Positional Cα root-mean-square fluctuation (RMSF) of (a) TEM-1 Apo$_{EQ}$, (b) TEM-1 IB$_{EQ}$, (c) KPC-2 Apo$_{EQ}$, and (d) KPC-2 IB$_{EQ}$ systems.

**Figure supplement 2.** Snapshot of the last frame from TEM-1 IB$_{EQ}$ and KPC-2 IB$_{EQ}$ replicate simulations.

**Figure supplement 3.** Average Cα deviation between the IB$_{EQ}$ and Apo$_{NE}$ calculated using the subtraction method for (a) TEM-1 and (b) KPC-2.

---

perturbation (ligand removal) show the route by which structural response is transmitted through the protein.

This approach has identified a general mechanism of signal propagation in nicotinic acetylcholine receptors, by analyzing their response to deletion of nicotine (*Oliveira et al., 2019a*). The difference in the position of Cα atoms is calculated between the short Apo$_{NE}$ and IB$_{EQ}$ simulations at specific time points. These differences are then averaged over all pairs of simulations to reveal the structural conformations associated with this response (*Figure 4*) and their statistical significance. The Cα coordinates of each residue in the Apo$_{NE}$ were subtracted from the corresponding Cα atom coordinates of the IB$_{EQ}$ simulation at specific points in time, namely 0.05, 0.5, 1, 3, and 5 ns. This resulted in a difference trajectory for each pair of simulations. The difference trajectories are averaged over the set of 800 simulations for each system. The low standard error (SE) calculated for the average between the Apo$_{NE}$ and IB$_{EQ}$ demonstrates the statistical significance of the results. Due to the short timescale (5 ns) of the nonequilibrium simulations, only small amplitude conformational changes will be observed.

In TEM-1, the allosteric site is sandwiched between the α11 and α12 helices. Adjacent to this binding site is the hinge region (residues 213–218), whose dynamics have previously been examined by NMR and shown to have low order parameters indicating high mobility (*Gobeil et al., 2019*; *Savard and Gagné, 2006*). This is also the site of perturbation in the nonequilibrium simulations and so the point of origin of the allosteric signal. Located on the loop between the distal end of the α11 helix and β7 is a highly conserved Trp229 residue. The indole ring of Trp229 is sandwiched between two other highly conserved residues, Pro226 and Pro251, present in loops α11-β7 and β8-β9, respectively. The π/aliphatic stacked arrangement of tryptophan-proline is a very tight interaction and is similar in geometry to that observed in complexes of proline-rich motif binding families, including the EVH1 and GYF binding domains, with their peptide ligands (*Ball et al., 2005*; *Freund et al., 1999*; *Reinhard et al., 1996*; *Zondlo, 2013*). The perturbation destabilizes this stacked arrangement resulting in an extension of an inherently highly mobile region. After 50 ps of simulation, the Cα deviations have propagated and can be observed in the loop between β1 and β2. Interestingly, the loops at the basal pivot of 3 and 4 also responded rapidly to ligand removal. These loops are ~33 Å away from the allosteric binding site and can affect the spatial position of the turn between helix α3 and α4. The α3-turn-α4 helix forms the boundary of the active site, and it is on this turn where the Tyr105 residue, important for substrate recognition, is positioned. These results clearly demonstrate the coupling between the distal allosteric site and catalytically relevant regions of the enzyme. As the signal propagates within the protein, there is a gradual and cumulative increase in the Cα deviations in the aforementioned loops. In particular, the loop between the α9 and α10 helices, which is positioned just below the β1-β2 loop, displays high deviations and forms a focal point for the signal to bifurcate in two directions. First, major deviations are observed laterally toward loop α7-α8 and onward into the Ω-loop (*Figure 4a,b*). Second, more minor deviations move into the loop between α2 and β4 and onward into the basal pivot of α3-turn-α4 helix. There is another shorter route at the top of the enzyme that the signal can take to go from the allosteric binding site to the Ω-loop, via the proximal end of α12 helix and across the loop between β9 and α12 helix (*Figure 4a,b*).

In KPC-2, the allosteric pocket is shallower and lies between helices α2 and α7. Residues from three loops (α6-α7, α7-α8, and α2-β4) are in close proximity to this binding site. An additional loop, α9-α10, is linked to this binding site via the distal end of the α2 helix. The perturbation in this binding site results in enhanced mobility of the α2-β4 loop, which leads directly into β4 and onward to the basal pivot of the α3 helix. The proximal end of the α3 helix and the distal end of the α4 helix,

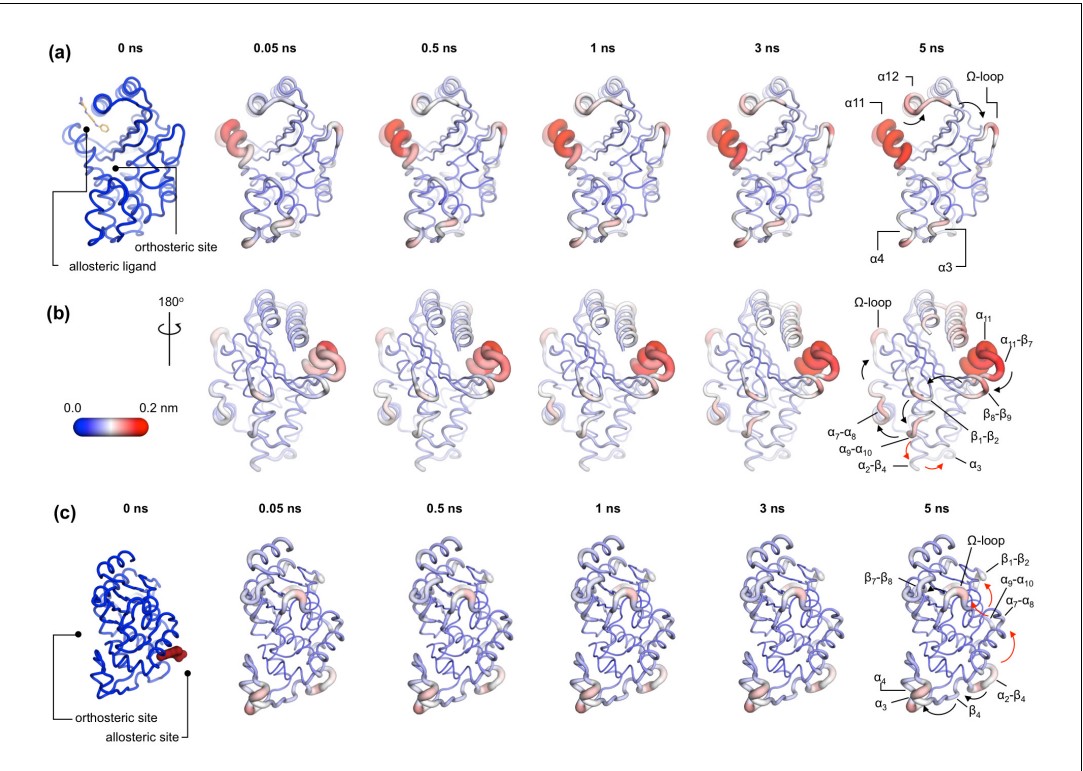

**Figure 4.** Communication pathways in (**a, b**) TEM-1 and (**c**) KPC-2. The average Cα deviations correspond to the average difference in the position of each Cα atom between all 800 pairs of IB$_{EQ}$ and Apo$_{NE}$ simulations at specific time points. The averaged Cα deviations are mapped onto the average Apo$_{EQ}$ structure. The arrows mark the direction of the propagation of the signal, caused by the perturbation (removal of the ligand). The red and the black arrows highlight different paths taken by the propagating signals (also see movies *Figure 4—video 1*, *Figure 4—video 2*, *Figure 4—video 3*).

The online version of this article includes the following video(s) for figure 4:

**Figure 4—video 1.** Signal propagation in TEM-1 (front view).

https://elifesciences.org/articles/66567#fig4video1

**Figure 4—video 2.** Signal propagation in TEM-1 (back view).

https://elifesciences.org/articles/66567#fig4video2

**Figure 4—video 3.** Signal propagation in KPC-2.

https://elifesciences.org/articles/66567#fig4video3

which forms the pivot point of the α3-turn-α4 structure, display high deviations (*Figure 4c*). The highly conserved aromatic amino acid, Trp105, is located on this turn. The distance between the allosteric binding site and the α3 helix is ~27 Å. Other major deviations are also observed in the Ω-loop as the simulation progresses (*Figure 4c*). The Ω-loop is directly linked to the allosteric binding site via loop α7-α8. Some minor deviations are also observed in the loop connecting β9 and the α12 helix.

In both TEM-1 and KPC-2, the removal of the ligand at the beginning of the nonequilibrium simulations does not result in large conformational changes. The subsequent Cα deviations trace the route of the propagating signals (*Figure 3—figure supplement 212*). In TEM-1, α11 and the hinge region, loops β1-β2 and β8-β9, respond rapidly to the perturbation and display comparable RMSD values to the equilibrated simulations. Similarly, in KPC-2, only loops α2-β4 and α7-α8 respond rapidly to the perturbation. The other structural elements take longer to respond, and their conformational rearrangements are not fully sampled in the Apo$_{NE}$ simulations. It is worth emphasizing that while the short nonequilibrium simulation can be an excellent tool to study an immediate structural response toward a perturbation, the timescale of nonequilibrium MD does not represent a real timescale and thus should not be compared directly with equilibrium simulations. Nevertheless, nonequilibrium MD can identify the sequence of events and pathways involved.

The perturbations of the two enzymes here are different but show some striking common features. In both TEM-1 and KPC-2 systems, even though the point of origin of perturbation (i.e. allosteric site) is different, the signal leads to common endpoints at the pivot of α3-turn-α4 helix and in the Ω-loop. Thus, simulations of two different class A β-lactamases, starting from two distinct allosteric sites, identify a common mechanism by which catalytic activity may be disrupted by conformational changes close to the active site. The results from the nonequilibrium simulations also correlate well with experimental data, which suggest that the Ω-loop plays a critical role in ligand binding by altering the conformation of Glu166 and Asn170 which are involved in both acylation and deacylation reactions (*Chudyk et al., 2014*; *Pan et al., 2017*; *Brown et al., 2009*; *Fritz et al., 2018*; *Banerjee et al., 1998*).

## Dynamic cross-correlation analysis of surface loops

Dynamical cross-correlation analysis provides information about the pathways of signal propagation and also some insights into the timescales of allosteric communication in TEM-1 and KPC-2 β-lactamases. Dynamic cross-correlation maps (DCCMs) have been previously used to identify networks of coupled residues in several enzymes (*Agarwal et al., 2004*; *Hester et al., 2019*; *Agarwal et al., 2012*).

Using a similar approach, DCCMs were calculated for the $Apo_{EQ}$ and $IB_{EQ}$ simulations and also for the $Apo_{NE}$ nonequilibrium simulations (*Figure 5*). In these figures, the green regions represent no to slightly positive correlations, while yellow regions represent moderate negative correlations. Negative correlations imply residues moving toward or away from each other in correlated fashion (such as shown by fluctuating hydrogen bonds); for large regions this represents global conformational fluctuations (also referred to as breathing motions) (*Agarwal et al., 2004*). The results depicted in *Figure 5a* indicate that in the case of TEM-1 $Apo_{EQ}$ (*Figure 5a*, left), β11 helix shows high negative correlation with β12 terminal helix. This represents the lid motion of β12 helix, which moves to shut the empty, hydrophobic, allosteric binding site in the TEM-1 $Apo_{EQ}$ structure (see above). This motion is, however, not observed in the ligand bound TEM-1 $IB_{EQ}$ simulations. The TEM-1 $IB_{EQ}$ system shows a substantial increase in correlations, representing changes in the dynamical communications due to the presence of the allosteric ligand (*Figure 5a*, middle). The binding of the ligand changes the overall global conformational fluctuations of TEM-1, as represented by the increase in yellow regions in the DCCMs. Furthermore, a number of negative correlations (encircled red regions in DCCMs) also increase in other regions of the protein on ligand binding. The DCCM collectively computed from all nonequilibrium trajectories for TEM-1 (*Figure 5a*, right, *Figure 5—figure supplement 1*) also shows a further increase in the areas of negative correlations (encircled). Interestingly, DCCM also identifies the pathway of allosteric communication (*Figure 5—figure supplement 1*), with notable correlations between the regions β1-β2:α2-β4, α3-α4:α2-β4, β4-α3:α7-α8, β3-α2:Ω, α9-α10:β1-β2, β3-α2:β8-β9, α5-α6:α12, hinge-α11:α1-β1, β8-β9:α4-β5, β7:α12, and α11:α12. These results indicate that the presence of ligand in TEM-1 increases the dynamic communication between regions that are independent in the $Apo_{EQ}$ simulations. This is particularly evident in the nonequilibrium trajectories that show the largest changes from the case of $Apo_{EQ}$ TEM-1, identifying changes in correlation as the system adjusts to the absence of the ligand.

KPC-2 shows even more interesting behavior (*Figure 5b*). Simulations of $Apo_{EQ}$ KPC-2 show overall more correlated regions than TEM-1 $Apo_{EQ}$ system (as indicated by the more extensive yellow regions in the DCCM), with further increases in the presence of the inhibitor (indicated by a number of orange regions). However, the DCCM collectively computed from all nonequilibrium trajectories for KPC-2 shows a reduction in regions of cross-correlations, a contrast from the case of TEM-1. To obtain a better understanding, the DCCMs from individual 5 ns nonequilibrium trajectories were also computed and analyzed. These reveal interesting trends as depicted in *Figure 5—figure supplement 2*. For most nonequilibrium trajectories, the maps are similar with a decrease in dynamic correlations; however, for several trajectories (shown in *Figure 5—figure supplement 2*), the maps indicate a significant increase in the correlations. The DCCMs computed from individual trajectories show behavior similar to averaged nonequilibrium trajectories in TEM-1 with a number of regions showing high negative correlations (as highlighted by widespread presence of small red regions in the DCCMs). Overall, these results indicate that the perturbation in KPC-2 generates a dynamical response that is much faster than that observed in TEM-1. A plausible explanation for the faster response in KPC-2 is that the more solvent-exposed ligand binding site is surrounded by dynamic

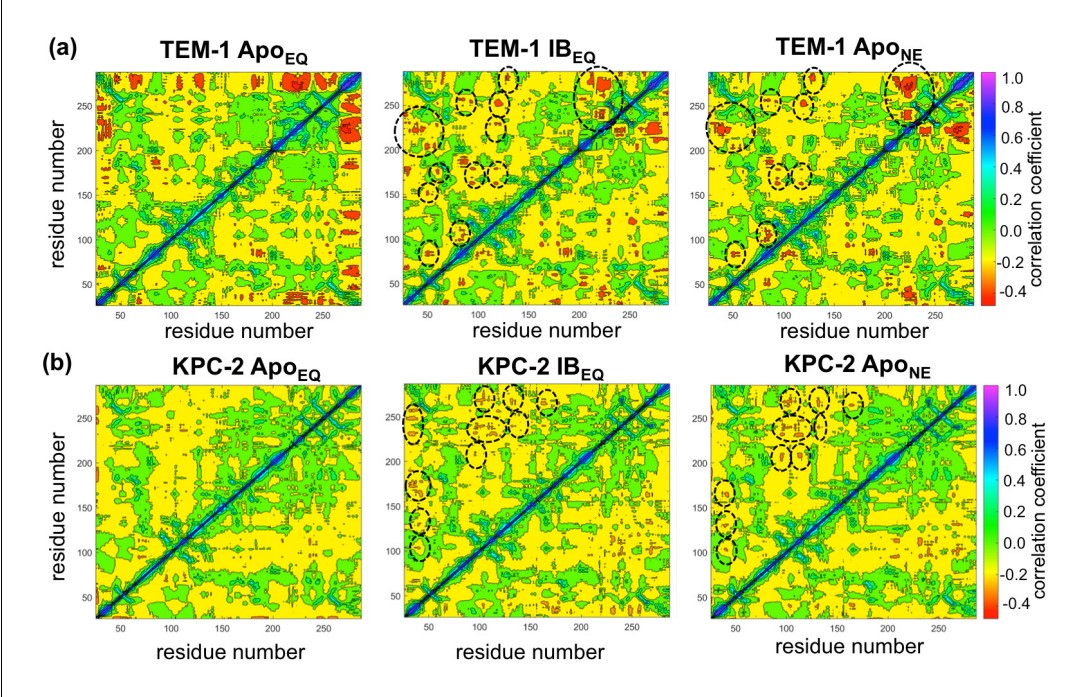

**Figure 5.** Dynamic cross-correlation maps (DCCMs) computed for (**a**) TEM-1 and (**b**) KPC-2 Apo equilibrium (Apo$_{EQ}$), inhibitor-bound equilibrium (IB$_{EQ}$), and Apo nonequilibrium (Apo$_{NE}$) trajectories. The DCCMs for equilibrium trajectories were calculated as an average of 20 replica simulations, while the Apo$_{NE}$ DCCM indicates an averaged DCCM from an ensemble of 40 short (5 ns) MD trajectories. Green regions indicate no correlation, yellow indicates moderate negative correlation, while orange and red indicate significant negative correlations and blue regions indicate positive correlations. In TEM-1 Apo$_{NE}$, regions showing significant changes from Apo$_{EQ}$ and IB$_{EQ}$ bound simulations have been marked by black dashed ellipses.

The online version of this article includes the following figure supplement(s) for figure 5:

**Figure supplement 1.** TEM-1 averaged dynamic cross-correlation map (DCCM) computed from all nonequilibrium trajectories.

**Figure supplement 2.** Selected dynamic cross-correlation maps (DCCMs) computed for individual 5 ns nonequilibrium molecular dynamics (MD) trajectories of KPC-2.

surface loops that respond to the perturbation more quickly than the allosteric binding site in TEM-1, which is buried in the hydrophobic core of the protein. This is consistent with the experimental observations that motions can occur on different timescales and can vary greatly between different β-lactamases (*Gobeil et al., 2019*).

## Relating enzyme dynamics to positions of substitution in TEM-1 and KPC-2 clinical variants

A number of clinical variants that extend hydrolytic activity to encompass additional β-lactams such as oxyiminocephalosporins, and/or enhance enzyme stability, have been identified for both the TEM-1 and KPC-2 β-lactamase enzymes (*Palzkill, 2018*; *Clark et al., 2016*; *Naas et al., 2017*). Some of these have been crystallized and their protein structure deposited in the PDB. While many of these amino acid substitutions (e.g. TEM-1 mutations at residues Glu104 in the α3-turn-α4, Arg164 on the Ω-loop, and Ala237, Gly238, and Glu240 on the β7 strand) directly affect important structural features such as the active site or the Ω-loop, some are of uncertain structural significance. Even when enzyme structures are known, the connections between the positions of clinical variants, protein structure, and their functional implications are often not clear. There is particular uncertainty and interest in the effects of mutations more distant from the active site.

To assess how many of these clinically relevant substitutions lie on the allosteric communication pathway, their spatial positions were identified and mapped onto the 3D structures of TEM-1 and KPC-2. The site of the mutation was plotted as a sphere on its unique Cα position on the structure (*Figure 6*), which was rendered to represent the allosteric communication pathways shown in *Figure 4*.

For TEM-1, 45 of the 90, and for KPC-2 15 out of the 25, amino acid positions known to vary in clinical isolates could be mapped onto the allosteric communication pathway. Notably, in TEM-1, residues such as Gly92 preceding α4, His153 at the end of α7, and Ala224 preceding α11 have all been associated with ESBL and/or inhibitor-resistant phenotypes identified in the clinic (*Palzkill, 2018*). Residues such as M182 and A184, which precede α9 and are not on the communication pathway per se, are however surrounded on all sides by loops that are involved in the communication network (*Figure 6—figure supplement 1*). For KPC enzymes, for which less information is available, characterized variants that have emerged in the clinic differ mostly in activity toward ceftazidime 58 and feature substitutions at positions (104, 240, 274) closer to the active site. As more sequences emerge and their phenotypic consequences are described (*Tooke et al., 2021*), however, it will then be of interest to establish the properties of KPC variants featuring substitutions at positions (e.g. 92, 93), which lie along the communication pathways described here. We propose that some of these variants differ in allosteric properties, and further, that these differences relate to variances in their clinically relevant spectrum of activity. If our hypothesis is correct, 50% or more of known clinically important variants in these two enzymes may differ in their allosteric behavior, indicating that this is a fundamentally important property in determining their spectrum of catalytic activity. The relationship between sequence (especially substitutions remote from the active site), protein dynamics, spectrum of activity, catalytic turnover, and allosteric behavior will be an important future direction in understanding AMR due to β-lactamase enzymes (*Tooke et al., 2021*).

## Discussion

Here, we have identified structural communication between two allosteric binding sites and structural elements, close to the active site, that control enzyme specificity and activity in two distinct, clinically important, class A β-lactamases. The extensive equilibrium MD simulations, with and without ligands, reveal ligand-induced conformational changes, while nonequilibrium MD simulations show that changes at allosteric sites are transmitted to the active site and identify the structural pathways involved. These nonequilibrium simulations identify the initial stages of the dynamic rearrangement of secondary structural elements and highlights the signal propagation routes (with

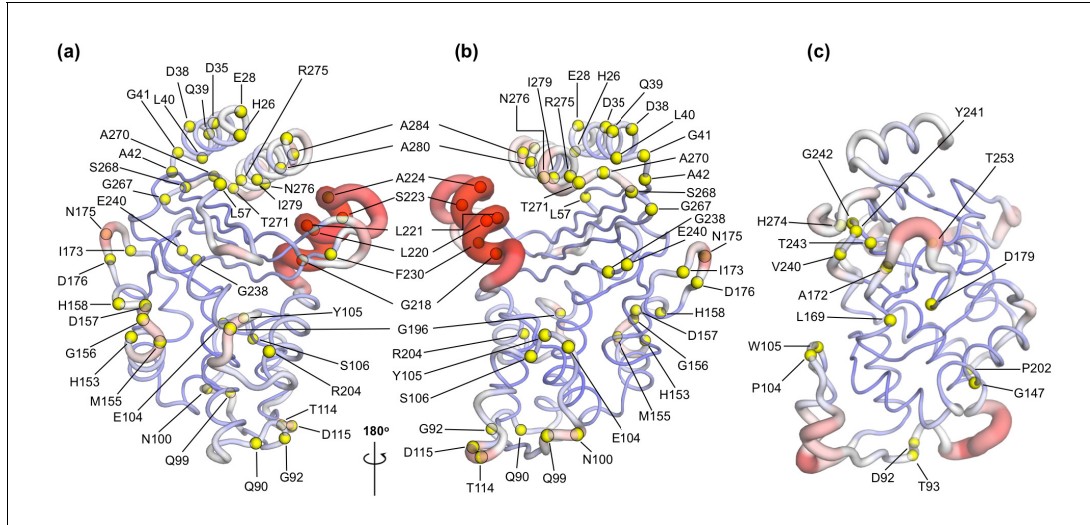

**Figure 6.** Variant positions in (a, b) TEM-1 and (c) KPC-2 mapped onto the averaged Apo_EQ structures, also showing allosteric communication pathways (see *Figure 4*) identified by nonequilibrium simulations. The position of the variant is shown as yellow spheres centered at the corresponding Cα. Only the sites of mutations that lie on the allosteric communication pathways have been annotated. The color scheme and cartoon thickness of the rendered structures represents a snapshot of average Cα deviation between IB_EQ and Apo_NE. Many of these clinically important variant positions lie on the allosteric communication pathway: 45 of the 90 for TEM-1, 15 out of the 25 for KPC-2 single point variants lie on the pathways. This suggests that these variations affect the allosteric behavior of the enzymes.

The online version of this article includes the following figure supplement(s) for figure 6:

**Figure supplement 1.** Spatial position of M182 and A184 on TEM-1.

demonstration of its statistical significance). These two complementary approaches together facilitate understanding of how information flows from one part of the protein structure to another.

The equilibrium simulations (of ligand-bound and Apo enzymes) show that the structural effects of ligand binding to allosteric sites are not restricted to the local binding pocket. Class A β-lactamases are rigid enzymes (*Gobeil et al., 2019*) that do not undergo large-scale conformational changes; the observed structural rearrangements (caused by ligand removal) are dominated by localized changes in the conformation of loops. Such ligand-induced structural changes are observed in the loops surrounding the active sites including the hinge region, the Ω-loop, and the α3-turn-α4 helix, positioned as far as ~33 Å from the allosteric ligand binding site. In both enzymes, the observed flexible motions lead to an enlargement of the active site, with the potential consequences for the orientation of either mechanistically important regions of the protein or of bound ligand, and, consequently, enzyme activity.

The nonequilibrium simulations, using an emerging technique, identify the structural rearrangements arising as a result of a perturbation (ligand removal) and demonstrate communication between the allosteric site and the active site. The ordering of these conformational changes shows the initial steps of communication between secondary structure elements. This structural relay constitutes a pathway that enables effective signal propagation within the enzymes. In TEM-1, the conformational changes initiated at the allosteric site (which is situated between helices α11 and α12) proceed via the β1-β2 loop to the α9-α10 loop. From this point, the signal bifurcates toward the Ω-loop via the α7-α8 loop or toward the α3-α4 pivot via the α2-β4 loop. In KPC-2, the perturbation caused by ligand unbinding between the α2 and α7 helices results in conformational changes in loop α2-β4, leading to β4 and onward to the pivot of the α3-turn-α4 helix. These conformational changes are relayed to the Ω loop via the α7-α8 loops. In addition, the signal can also take another route from the α7-α8 loop toward the β9-α12 loop, which lies adjacent to the hinge region. It is worth emphasizing that the TEM-1 and KPC-2 systems display a striking resemblance in that the flow of information is toward a common endpoint, despite the two different points of origin. Thus, even though the propagation pathway taken is different, in each case, the signals accumulate to have a structural impact on the conformation of the Ω loop and the α3-turn-α4 helix. These results demonstrate communication between allosteric ligand binding sites and the active sites of the enzymes, which could be exploited in alternative strategies for inhibitor development.

All class A β-lactamase enzymes share conserved structural architecture (*Philippon et al., 2016*; *Galdadas et al., 2018*). Mutational studies and the location of sites of substitutions in clinical variants suggest the importance to activity of the hinge region, Ω-loop, and α3-turn-α4 helix, including the spatial position of the conserved aromatic residue at 105 (or the analogous position in other class A β-lactamases) (*Palzkill, 2018*; *Philippon et al., 2016*; *Banerjee et al., 1998*; *Papp-Wallace et al., 2010b*). Perturbations around these sites, as identified in the simulations here, may constitute a general mechanism by which a conformational signal transmitted from an allosteric site is relayed via cooperative coupling of loop dynamics to affect catalytic activity. Exploitation of such signaling networks may constitute a novel strategy for the development of new types of inhibitors for these key determinants of bacterial antibiotic resistance.

## Materials and methods

### Protein structure preparation

To study allosteric modulation of class A β-lactamases, we started by identifying crystal structures of TEM-1 and KPC-2 β-lactamases with allosteric ligands bound. From the ~80 structures present in the PDB, there are only two crystal structures of class A β-lactamases that have a ligand bound in an allosteric pocket. For TEM-1, the 1.45 Å crystal structure in complex with FTA [3-(4-phenylamino-phenyl-amino)−2-(1h-tetrazol5-yl)-acrylonitrile] was chosen as the starting structure (PDB id: 1PZP) for this work *Horn and Shoichet, 2004*. In this structure, the inhibitor binds between helices α11 and α12 (*Figure 1a*), in a site ~16 Å away from the active site Ser70. Two unstructured residues from the C-terminal end (His289, Trp290) were removed from the crystal structure. For KPC-2, the 1.35 Å crystal structure in complex with a coumarin phosphonate analogue, GTV [(5,7-dimethyl-2-oxo-2h-1-benzopyran-4-yl)methylphosphonic acid], was chosen as the starting structure (PDB id: 6D18) (*Pemberton et al., 2019*). GTV binds in three sites on KPC-2 (*Figure 1b*): the first is in the active site

(orthosteric ligand); the second site is adjacent to helix α6 (allosteric ligand 1); and the third (allosteric ligand 2) is on the distal end of the enzyme, ~16 Å from the active site Ser70 in between helices α2 and α7. The orthosteric and allosteric ligand 1 (*Figure 1b*) were discarded because of their direct proximity to the active site and replaced by water. Three unstructured residues from the N-terminal end (His23, Met24, Leu25) and seven from the C-terminal end (Leu288-Gly294) were removed from the starting structure to avoid any simulation artifacts arising as a result of terminal fraying during simulations.

The protonation states of the amino acid side chains were determined at pH 7.0, using the *ProteinPrepare* functionality as implemented in the high-throughput molecular dynamics (HTMD) framework (*Martínez, 2015*; *Doerr et al., 2016*). Charges were assigned on the basis of their local environment, via optimization of the hydrogen-bonding network of the protonated structure (*Martínez, 2015*).

Parameters for the ligands were generated using the Antechamber tool (*Case et al., 2005*). The geometry was optimized at the B3LYP/6-31G(d) level and RESP charges were fitted using electrostatic potential obtained at the HF/6-31G(d) level. The necessary nonbonded parameters for the dynamics of the ligands were adopted from GAFF2 (*Wang et al., 2004*).

## MD simulations details

All complexes were set up using tleap, as implemented in the Amber MD package. The Amber ff14SB forcefield (*Maier et al., 2015*) was used for the protein. In total, four complexes were set up, including an allosteric IB (inhibitor-bound) and an Apo (no ligand) system for both TEM-1 and KPC-2 β-lactamases. The Apo system was generated by removing the inhibitor from the allosteric binding site. In all simulated complexes, there is no ligand bound to the orthosteric site. Each complex was solvated using TIP3P water in a cubic box, whose edge was set to at least 10 Å from the closest solute atom (*Mark and Nilsson, 2001*). The systems were neutralized using $K^+$ and $Cl^-$ counter ions. The simulation protocol was identical for each system. The systems were minimized and relaxed under NPT conditions for 5 ns at 1 atm. The temperature was increased to 300 K using a time step of 4 fs, rigid bonds and a cutoff of 9 Å, and particle mesh Ewald summations switched on for long-range electrostatics (*Essmann et al., 1995*). During the equilibration step, the protein's backbone and the ligand atoms were restrained by a spring constant set at 1 kcal mol$^{-1}$ Å$^{-2}$, while the ions and solvent were free to move. The production simulations were run in the NVT ensemble using a Langevin thermostat with a damping constant of 0.1 ps and hydrogen mass repartitioning scheme to achieve a time step of 4 fs (*Feenstra et al., 1999*). The final production step was run without any restraints. All simulations were run using the ACEMD MD engine as implemented in the HTMD framework (*Doerr et al., 2016*). Visualization of the simulations was done using the VMD package (*Humphrey et al., 1996*).

### Equilibrium simulations

In order to sufficiently sample the conformational space, 20 replicate simulations of 250 ns each were performed for each system. This resulted in a total sampling time of 5 μs for each system. The initial velocities of the atoms of each replica were randomized. We describe this set of runs in this study as equilibrium simulations (Apo$_{EQ}$/IB$_{EQ}$).

### Nonequilibrium simulations

To investigate rapid conformational changes and study signal propagation within the proteins, we carried out 800 short nonequilibrium MD simulations for each system. Such nonequilibrium simulations have been applied successfully to study interdomain communication in receptors and other systems like ABC transporters (*Abreu et al., 2020*; *Damas et al., 2011*; *Oliveira et al., 2019a*; *Oliveira et al., 2019b*; *Oliveira et al., 2005*). We used the Kubo-Onsager approach (*Ciccotti and Ferrario, 2013*; *Ciccotti and Ferrario, 2016*; *Ciccotti et al., 1979*) to extract the conformational response of the proteins to ligand removal. In this approach, the response of a system to a perturbation is computed by calculating the difference in the evolution of the simulations with and without the perturbation. Subtracting the perturbed and unperturbed pairs of simulations at a given time, and averaging the results over multiple replicates, allows not only for the identification of the events associated with signal propagation but also determines the statistical significance of the

observations. When the two sets of simulations (with and without a perturbation) are correlated, the subtraction technique permits the cancelation of noise arising from random intrinsic fluctuations of the system thus allowing the identification of the response to the perturbation in a statistically significant way (*Ciccotti et al., 1979*). In our systems, the perturbation was generated by (instantaneously) removing the ligand from the allosteric pocket. It is important to emphasize that the annihilation of the ligand in this way does not represent the physical process of unbinding. The objective is to rapidly elicit response and force signal propagation within the protein, as the conformation adjusts to the removal of the ligand. Such a response allows for the identification of the initial signals that are sent out as conformational changes associated with the signal propagating from the allosteric binding pocket. The structural rearrangements in the communication pathways revealed by nonequilibrium simulations are likely to be involved in response to the physical process of binding and unbinding of ligands in the allosteric pockets.

A graphical representation of the procedure that was followed to set up the nonequilibrium simulations is given in *Figure 2—figure supplement 1*. The starting conformation for the short nonequilibrium simulations (Apo$_{NE}$) was extracted from the equilibrated part of the 250 ns equilibrium simulations (50–250 ns). Specifically, conformations were taken every 5 ns, the ligand was removed from the allosteric pocket, and the resulting Apo$_{NE}$ system was run for another 5 ns (*Figure 2—figure supplement 1*). Forty short, nonequilibrium simulations were run for each replicate. In total, 800 simulations were run for each system. The simulation conditions of the nonequilibrium simulations were identical to those in the equilibrium simulations.

For each pair of unperturbed IB$_{EQ}$ and perturbed Apo$_{NE}$ simulations, the difference in positions for each C$\alpha$ was determined at equivalent points in time, namely at 0, 0.05, 0.5, 1, 3, and 5 ns. Calculating the differences in the positions of C$\alpha$ identifies conformational rearrangements, while reducing the noise coming from side chain fluctuations. The C$\alpha$ deviation values at each time point were averaged over all 800 simulations. To assess the statistical significance of the conformational response over hundreds of simulations performed, the standard deviation (SD) and SE of the mean (95% confidence interval) were determined. Overall, low SD and SE values observed for all the regions of interest (as illustrated for, e.g., in *Figure 3*) demonstrate the statistical significance of the results.

## Analysis details

The analysis was carried out using GROMACS tools (*Abraham et al., 2015*), MDLovofit (*Martínez, 2015*), and in-house scripts (*Oliveira et al., 2019a*). All systems were considered equilibrated after 50 ns. The dynamic cross-correlations for C$\alpha$-C$\alpha$ were calculated using cpptraj analysis program (*Roe and Cheatham, 2013*). The results were plotted using in-house scripts and visualized using MATLAB (http://www.mathworks.com).

An independent-samples Student's t-test was used to compare the Apo$_{EQ}$ and IB$_{EQ}$ RMSFs and to assess the significance of the differences observed (*Oliveira et al., 2019a*; *Roy and Laughton, 2010*). The sample size used for the t-test was the 20 RMSF profiles of the Apo$_{EQ}$ and IB$_{EQ}$ independent simulations. The assumption used for the t-test was that the samples from the two states were independent, the dependent variable was normally distributed, and the variances of the dependent variable were equal.

The figures were made using PyMol (http://www.schrodinger.com), VMD (*Humphrey et al., 1996*), ChimeraX (*Goddard et al., 2018*), Protein Imager (3dproteinimaging.com) (*Tomasello et al., 2020*), and Molsoft ICM-Pro package (http://www.molsoft.com).

## Acknowledgements

IG is funded by Astra Zeneca-EPSRC case studentship awarded to FLG and SH. PKA acknowledges a grant from the National Institute of General Medical Sciences of the National Institutes of Health USA under award number GM105978. SH and RB acknowledge a grant from the National Institutes of Health USA under the award number RO1AI063517. AJM and ASF thank EPSRC for support (grant numbers EP/M022609/1 and EP/N024117/1) and also thank BrisSynBio, a BBSRC/EPSRC Synthetic Biology Research Centre (grant number BB/L01386X/1) for funding. AJM, JS, and CLT also thank MRC for support (grant number MR/T016035/1). RAB is supported by the National Institute of Allergy and Infectious Diseases of the National Institutes of Health (NIH) under Award Numbers

R01AI100560, R01AI063517, and R01AI072219 and in part by funds and/or facilities provided by the Cleveland Department of Veterans Affairs, Award Number 1I01B × 001974 from the Biomedical Laboratory Research and Development Service of the VA Office of Research and Development, and the Geriatric Research Education and Clinical Center VISN 10. The content is solely the responsibility of the authors and does not necessarily represent the official views of the NIH or the Department of Veterans Affairs.

## Additional information

### Competing interests

Pratul K Agarwal: Pratul K Agarwal is the founder and owner of Arium BioLabs LLC. The other authors declare that no competing interests exist.

### Funding

| Funder | Grant reference number | Author |
|---|---|---|
| AstraZeneca | Case Studentship | Ioannis Galdadas |
| National Institute of General Medical Sciences | GM105978 | Pratul K Agarwal |
| National Institutes of Health | RO1AI063517 | Robert A Bonomo Shozeb Haider |
| Engineering and Physical Sciences Research Council | EP/M022609/1 | Ana Sofia F Oliveira Adrian J Mulholland |
| Engineering and Physical Sciences Research Council | EP/N024117/1 | Ana Sofia F Oliveira Adrian J Mulholland |
| Biotechnology and Biological Sciences Research Council | BB/L01386X/1 | Ana Sofia F Oliveira Adrian J Mulholland |
| Medical Research Council | MR/T016035/1 | Catherine L Tooke James Spencer Adrian J Mulholland |
| National Institute of Allergy and Infectious Diseases | R01AI100560 | Robert A Bonomo |
| National Institute of Allergy and Infectious Diseases | R01AI063517 | Robert A Bonomo |
| National Institute of Allergy and Infectious Diseases | R01AI072219 | Robert A Bonomo |

The funders had no role in study design, data collection and interpretation, or the decision to submit the work for publication.

### Author contributions

Ioannis Galdadas, Formal analysis, Investigation, Visualization, Writing - review and editing; Shen Qu, Formal analysis, Investigation; Ana Sofia F Oliveira, Software, Supervision, Validation, Methodology, Writing - review and editing; Edgar Olehnovics, Investigation, Visualization; Andrew R Mack, Maria F Mojica, Catherine L Tooke, Formal analysis, Investigation, Writing - original draft; Pratul K Agarwal, Formal analysis, Investigation, Writing - original draft, Writing - review and editing; Francesco Luigi Gervasio, Robert A Bonomo, Supervision, Funding acquisition, Writing - review and editing; James Spencer, Formal analysis, Supervision, Investigation, Writing - review and editing; Adrian J Mulholland, Conceptualization, Supervision, Methodology, Writing - original draft, Writing - review and editing; Shozeb Haider, Conceptualization, Data curation, Formal analysis, Supervision, Funding acquisition, Validation, Investigation, Visualization, Methodology, Writing - original draft, Project administration, Writing - review and editing

### Author ORCIDs

Ioannis Galdadas [iD] https://orcid.org/0000-0003-2136-9723
Andrew R Mack [iD] https://orcid.org/0000-0002-0131-7996
Francesco Luigi Gervasio [iD] http://orcid.org/0000-0003-4831-5039
Adrian J Mulholland [iD] http://orcid.org/0000-0003-1015-4567
Shozeb Haider [iD] https://orcid.org/0000-0003-2650-2925

## Decision letter and Author response

Decision letter https://doi.org/10.7554/eLife.66567.sa1
Author response https://doi.org/10.7554/eLife.66567.sa2

## Additional files

### Supplementary files

• Transparent reporting form

### Data availability

All analysis scripts have been uploaded on figshare with https://doi.org/10.6084/m9.figshare.13583384.

The following dataset was generated:

| Author(s) | Year | Dataset title | Dataset URL | Database and Identifier |
|---|---|---|---|---|
| Haider S | 2020 | Non-equilibrium simulation scripts | https://doi.org/10.6084/m9.figshare.13583384 | figshare, 10.6084/m9.figshare.13583384 |

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
