## [Decision Letter]

**Acceptance summary:**

This manuscript presented an extensive computational study aimed at deciphering the allosteric signaling propagation pathway in two class-A β-lactamases. The results of this study will be of interest to the readers in the field of β-lactamase, antibiotic resistance, and enzyme allostery.

**Decision letter after peer review:**

Thank you for submitting your article "Allosteric communication in class A β-lactamases occurs via cooperative coupling of loop dynamics" for consideration by *eLife*. Your article has been reviewed by three peer reviewers, including Yogesh K Gupta as the Reviewing Editor and Reviewer #1, and the evaluation has been overseen by Mone Zaidi as the Senior Editor. The following individuals involved in review of your submission have agreed to reveal their identity: Yun Lyna Luo (Reviewer #2); Bernard Fongang (Reviewer #3).

Essential Revisions:

Please address a few questions raised by the reviewer #3. The comments are appended.

Reviewer #3 (Recommendations for the authors):

The manuscript's language is appropriate with almost no typos (Except for “The systems were…”)

Subsection “Nonequilibrium simulations”: It is not exactly clear how subtracting the perturbed and the unperturbed pairs of simulations would help determine the statistical significance of the observations. What would be the threshold above which the observations are significant? A better explanation of the statistical significance would also help explain your conclusions.

Although you indicated all the software used for the Molecular Dynamics simulations and the visualization, I believe that sharing your codes and in-house scripts will benefit other researchers in the field.

Are you considering sharing your scripts with the scientific community?

---

## [Author Response]

Reviewer #3 (Recommendations for the authors):The manuscript's language is appropriate with almost no typos (Except for “The systems were…”)Subsection “Nonequilibrium simulations”: It is not exactly clear how subtracting the perturbed and the unperturbed pairs of simulations would help determine the statistical significance of the observations. What would be the threshold above which the observations are significant? A better explanation of the statistical significance would also help explain your conclusions.

The immediate response of a system to a perturbation is directly measured by averaging the difference of a given property (in this case, the positions of the C-α atoms) in nonequilibrium and equilibrium simulations at equivalent points in time as long as the two simulations are highly correlated and providing that enough data is gathered. In highly correlated systems, the random fluctuations of the systems largely cancel, thus giving the time evolution of the response to the perturbation.

For long simulation times, and as the correlation between the trajectories is lost, the subtraction technique is no longer useful (Ciccotti et al., 1979). Thus this method allows the identification of the first conformational changes associated with signal propagation through identification of common features. To assess the statistical significance of the conformational response over the hundreds of simulations performed, the standard deviation (SD) and standard error (SE) of the mean (95% confidence interval) were determined. Overall, the low SD and SE values observed for all the regions of interest (as illustrated for e.g. in Figure 3) demonstrate the statistical significance of the results. This has now been included in the main text.

The equilibrium and nonequilibrium MD simulations performed here are complementary approaches to study signal propagation and the communication between domains. The equilibrium simulations allow for the determination of the conformational rearrangements induced removal of ligand, after hundreds of nanoseconds of simulation. On the other hand, the nonequilibrium simulations with the use of the Kubo-Onsager approach, allow identification of the initial steps of the conformational change and reveal communication between the allosteric site and the active site. They reveal the structural mechanism of this coupling, in a statistically significant manner (due to hundreds of simulations performed). Nevertheless, we need to keep in mind that due to the short timescale of these nonequilibrium simulations, the observed structural rearrangements only reflect the first steps involved in signal propagation. Rearrangements taking longer are not sampled (bear in mind that the timescales in the nonequilibrium simulations are not the timescales of the conformational changes as would be physically associated with ligand binding or dissociation).

Although you indicated all the software used for the Molecular Dynamics simulations and the visualization, I believe that sharing your codes and in-house scripts will benefit other researchers in the field.Are you considering sharing your scripts with the scientific community?

We have uploaded our scripts to figshare and they can be downloaded from 10.6084/m9.figshare.13583384 We hope that these will be useful, and ask any users to cite our relevant work. We stress that the scripts are not generic (they are specific for the enzyme simulations here) and would need to be adapted for the system under study.